# Mallows Models for Top-$k$ Lists

**Flavio Chierichetti**
Sapienza University, Rome, Italy
flavio@di.uniroma1.it

**Anirban Dasgupta**
IIT, Gandhinagar, India
anirban.dasgupta@gmail.com

**Shahrzad Haddadan**
Sapienza University, Rome, Italy
shahrzad.haddadan@uniroma1.it

**Ravi Kumar**
Google, Mountain View, CA
ravi.k53@gmail.com

**Silvio Lattanzi**
Google, Zurich, Switzerland
silviolat@gmail.com

## Abstract

The classic Mallows model is a widely-used tool to realize distributions on permutations. Motivated by common practical situations, in this paper, we generalize Mallows to model distributions on top-$k$ lists by using a suitable distance measure between top-$k$ lists. Unlike many earlier works, our model is both analytically tractable and computationally efficient. We demonstrate this by studying two basic problems in this model, namely, sampling and reconstruction, from both algorithmic and experimental points of view.

## 1 Introduction

Ordering objects according to a set of criteria and presenting a prefix of the ordering to a user has become an accepted form of processed knowledge and is ubiquitous in practical settings. A search engine's result page typically contains the top ten results for a query. Newspapers and media outlets often list the top few movies in a year and the top few restaurants in a neighborhood. Social networking sites list the top followers or friends of a user. The popularity of top lists has resulted in the proliferation of dedicated portals such as listverse.com and www.thetoptens.com/lists.

Presenting a top list instead of a total ordering (permutation) also has many advantages in practice. First, the universe of objects might be too large to order and present, e.g., even for a niche subject like number theory, Google returns 9.7M Web pages. Secondly, the cognitive load on the user can become immense if the entire ordering is shown to her, especially when the most interesting piece of information is in a short prefix—in most cases, the user is indifferent to the 100th most popular restaurant in a city. Thirdly, it may be impossible or meaningless to total order the objects beyond a certain prefix length. Sociologists would be hard-pressed to pinpoint the 10000th most livable city.

Permutations and generative models for permutations have been studied for many decades, backed by a rich and elegant mathematical theory. In particular, the well-studied Mallows model offers the conceptual equivalent of the Gaussian distribution in the space of permutations: given a center permutation and a spread parameter, this generative model induces a distribution on all permutations, where the probability of a given permutation is a function of the (Kendall) distance to the center, scaled by the spread parameter. The problems of generating a random sample from a Mallows distribution [10], learning the center given a set of samples from the model [4, 5], and learning in a Mallows mixture setting [1, 6], have all been extensively studied in the literature. The ease of

analyzing the *repeated insertion model* (RIM) [10], the canonical way of generating a sample from Mallows distribution, forms a bedrock of such algorithms and their analysis.

The story, however, is far less mature for the study of top-$k$ lists. While top-$k$ lists have long been considered in the data mining, information retrieval, and machine learning communities, most of the work (with the exception of a handful of papers), has been mostly experimental [13, 18, 26, 24, 8, 12, 23]. In particular, there is no crisp extension of the Mallows model to the top list case with provable guarantees on the complexity of either sampling or for the reconstruction of the parameters. Any such model has to satisfy the following reasonable desiderata: the model should be conceptually simple, be a true generalization of Mallows, and the algorithms based on the model should have running times (and sample complexity for learning) that are polynomial only in the size of the top list rather than the entire universe. A reasonable and well-studied alternative, pioneered by [13] and developed since then [18, 23] (see related work for a longer discussion), is to posit the existence of a Mallows distribution over permutations of all the elements and yet regard only a prefix as the observable list. This, however, runs into the following issues: (i) generating a single sample from such distributions takes time polynomial in universe size, essentially because of using the RIM, (ii) semantically, even positing the existence of a full ranking and trying to learn it is potentially working with overspecified models, and hence, (iii) given a sample of top-$k$ lists that is of size polynomial in $k$, most of the pairwise relations that are learned are potentially statistically spurious (i.e., some pairs of items might be rarely observed). To alleviate these issues and to satisfy our desiderata, we first need a category of models that can replace Mallows models in the study of prefix lists.

**Our contributions.** A natural pathway towards such a model would be to consider extensions of the Kendall distance to top-$k$ lists. This is the route we take in this paper. We consider parametrized generalizations of Kendall distance to top-$k$ lists [11, 7, 16], which have nice mathematical properties, and use them to define a Mallows-like model for top-$k$ lists over universe of size $n$: given a top-$k$ center list and a spread parameter, our model induces a distribution on all top-$k$ lists.

While our top-$k$ Mallows model is easy to state, it poses several computational challenges, especially if we desire efficiency. We first consider the problem of generating a sample from this model. This problem, solved by RIM in all previous works, turns out to be non-trivial. We show two efficient algorithms: an exact sampling algorithm that runs in time $O(k^2 4^k + k^2 \log n)$ and a Markov chain-based approximate sampling algorithm that runs in time $O(k^5 \log k)$. We then consider two learning problems, namely, reconstructing the center for a given top-$k$ Mallows model and learning a mixture of top-$k$ Mallows models. For both these problems, we propose simple algorithms that have efficient sample complexities and provable guarantees. We also conduct experiments on both synthetic and real-world datasets to demonstrate the efficiency and usability of our algorithms in practice.

Our work (see Section 2 for formal definitions) differs from all previous works [13, 18, 23] on modeling top-$k$ lists in four aspects. Firstly, ours only posits the existence of a top-$k$ list and is based on a true generalization of Kendall distance to top-$k$ lists. Secondly, by using the $p$-parametrized Kendall distance, we control the probability of elements not in the center appearing in the generated top-$k$ list by changing the value of $p$, which is more general than specific distances [23].[1] Thirdly, in our model the probability of a pair of elements outside the center appearing in a generated top-$k$ list is *independent* of their relative order, whereas in previous works these probabilities are affected if this pair is inverted. As we discussed earlier, interpreting the ordering among elements outside the center is not meaningful in many applications; in this sense, our model is more natural and useful since it is able to control and minimize the influence of elements outside the center. Finally, our work focuses on the algorithmic questions with provable performance guarantees for both sampling and reconstruction, while previous works focused more on the statistical aspects of the model.

A criticism of our model could be that it treats all the tail elements equally. While this may look simplistic, assuming there is an implicit full ranking is unrealistic and more so for large domains. In this regard, our model offers a spectrum of flexibility. Firstly, note that $k$ is parameter and hence one can decide how far to dig into the tail for large domains. Secondly, in terms of fitting to observed data, our model has size $O(k \log n)$, rather than the $\Omega(n \log n)$ needed for the full permutation assumption, and hence, as mentioned before, is less prone to learning spurious orderings among most pairs of elements. Thirdly, our model admits natural generalizations in two ways. (i) One can define a general

top-$k$ model where the bottom $n − k$ elements are grouped into a small number of classes, each containing elements that are equally likely to appear in the top $k$. (ii) One can define a model where the center is a full permutation and a suitable Kendall-style distance function between a permutation and a top-$k$ list is chosen (e.g., an appropriate generalization of $K^{(p)}$ [11] or a Hausdorff variant [7]). A number of our results can be extended to both these generalizations (e.g., the exact/approximate sampling methods for (i)), or at least helps identify the right algorithmic tools (e.g., random walks and dynamic programming) to focus on for these extensions, while an RIM-based analysis cannot.

**Related work.** The papers closest to this work are [13, 18, 23]. Fligner and Verducci [13] define a distribution on partial rankings by taking a marginal over all possible extensions of the top-$k$ ranking. This idea is extended by Lebanon and Mao [18] to partial rankings, and also by giving a kernel density-based estimator for the underlying hidden permutation. While such an approach has the nice property of a closed-form expression for the sampling probability for top-$k$ lists (and some generalizations of that), it is not based on any distance function between top-$k$ lists, instead positing the existence of an underlying full permutation. Meila and Bao [23] propose a top-$k$ list generative model derived from [13] by considering the central permutation to be infinitely long. Their model is based on a specific set distance between a top-$k$ list and an permutation and assumes the existence of a full (infinitely long) permutation.

A RIM-like generative process and a reconstruction algorithm for a generalization of Mallows model using the 'average precision' distance which places more importance on the displacement of the top elements was proposed in [8] . Although the focus of their work is on full permutations, [20] outline a strategy to sample from any distribution over rankings as long as it is easy to sample from the corresponding conditional insertion probabilities; they show that this is the case for restricted classes of preference rankings that are generalizations of Mallows. Recently, [12] proposed a rank-dependent coarsening model for generating partial rankings and showed consistency of certain estimations. A number of other heuristics for estimating a consensus ranking by aggregation of partial rankings can be found in [7, 21].

Comparing top-$k$ lists has found several recent applications in comparing different measures of user importance in social networks [17, 27], diversifying recommendations [28], and various other information aggregation tasks [15, 14, 9].

## 2 Preliminaries

Let $[n] = \{1, \ldots, n\}$ be a universe of $n$ elements. A *top-k list* over $[n]$ is a partial order of the form $i_1 > \cdots > i_k > \{i_{k+1}, \ldots, i_n\}$ where $i_\ell$'s are elements of $[n]$. In other words, a top-$k$ list has $k$ elements that are strictly ordered and ranked above the remaining $n − k$ elements that are incomparable to each other. Let $T_{k,n}$ be the set of all top-$k$ lists over $[n]$; clearly $T_{n,n} = S_n$, the symmetric group on $n$ elements.

Throughout the paper, let $k \leq n$; we will think of $k$ as $O(\log n)$ or as a constant. We will use $\tau$ and subscripted versions to denote a generic top-$k$ list. Given $\tau \in T_{k,n}$ and an element $i \in [n]$, we use $i \in \tau$ to denote that $i$ is one of the top $k$ elements in $\tau$. For a position $i \in [k]$, we let $\tau(i)$ to denote the element at position $i$ in $\tau$. For $i, j \in [n]$, we use $i >_\tau j$ to denote that $i$ is ranked above $j$ in $\tau$, i.e., $i \in \tau$ and either $j \notin \tau$ or $j \in \tau$ but ranked below $i$. We use $i \parallel_\tau j$ to denote that $i \notin \tau$ and $j \notin \tau$, i.e., $i$ and $j$ are incomparable, and use $i \perp_\tau j$ to denote that $i$ and $j$ are comparable, i.e, either $i <_\tau j$ or $i >_\tau j$. Let $\bar{\tau} = [n] \setminus \tau = \{i \in [n] \text{ and } i \notin \tau\}$, the elements ranked below $\tau$. For a subset $S \subseteq [n]$, let $S \cap \tau = \{i \in \tau \text{ and } i \in S\}$.

We will consider the following distance measure [7, 11] that is a generalization of the well-known Kendall distance between permutations and while not a metric, has nice mathematical properties [11]. Let $p \geq 0$ be a parameter. The *p-parametrized Kendall distance* between $\tau_1, \tau_2 \in T_{k,n}$ is given by

$$K^{(p)}(\tau_1, \tau_2) = \sum_{i,j \in \tau_1 \cup \tau_2 \text{ and } i<j} \bar{K}_{i,j}^{(p)}(\tau_1, \tau_2),$$

where,

$$\bar{K}_{i,j}^{(p)}(\tau_1, \tau_2) = \begin{cases} 1 & (i <_{\tau_1} j \text{ and } i >_{\tau_2} j) \text{ or vice versa} \\ p & (i \perp_{\tau_1} j \text{ and } i \parallel_{\tau_2} j) \text{ or vice versa} \\ 0 & \text{otherwise.} \end{cases}$$

We next define the Mallows model for top-$k$ lists. Given a *center* top-$k$ list $\tau^*$ and a *decay* parameter $\beta > 0$, the *top-k Mallows model* induces a distribution on $T_{k,n}$ such that

$$\Pr[\tau \in T_{k,n}] \propto \exp\left(-\beta \cdot K^{(p)}(\tau^*, \tau)\right).$$

By relabeling, we can assume without loss of generality that $\tau^*$ is the "identity", i.e, $\tau^* = I_k \overset{\Delta}{=} 1 > \cdots > k > \{k+1, \ldots, n\}$. We denote $K^{(p)}(I_k, \tau)$ by $K^{(p)}(\tau)$. We refer to the above distribution as $\mathcal{M}^{(p)}_{\beta,\tau^*,k,n}$, abbreviating as $\mathcal{M}_{\beta,\tau^*}$ if $p$, $k$, $n$ are clear from the context and as $\mathcal{M}_\beta$ if $\tau^* = I_k$. Let

$$Z^{(p)}_{\beta,k,n} = \sum_{\tau \in T_{k,n}} \exp\left(-\beta \cdot K^{(p)}(\tau)\right),$$

be the normalizing constant. All the missing proofs are in the supplementary material.

## 3   Efficient sampling

We first study the problem of efficiently sampling from the top-$k$ Mallows model. Our goal is to obtain algorithms that are time- and space-efficient in terms of both $n$ and $k$, ideally, $O(n \cdot \text{poly}(k))$.

Recall that it is possible to efficiently sample the Mallows model on full permutations ($k = n$) using RIM [10]. A naive way of constructing a top-$k$ list would be to sample a full permutation according to RIM and discard the bottom $n - k$ elements; however, this process is incorrect, for instance, it does not account for the parameter $p$. A more direct approach would be to modify RIM to work in the top-$k$ case. To recap, in RIM, new elements are inserted one-by-one to construct a sample permutation, where at the $i$th step the element $i$ is inserted into the partial permutation $\sigma_1 \ldots \sigma_{i-1}$ at position $j \leq i$ with probability $p_{ij}$; it turns out that these insertion probabilities can be explicitly computed for a given parameter $\beta$ making the method computationally efficient. Unfortunately it is unclear how to adapt RIM to the top-$k$ setting for two reasons: (i) the insertion probabilities do not seem explicitly computable and (ii) the distance between two top-$k$ lists can decrease after the insertion of an element (note this is not true for full permutations).

In the following we present two algorithms for sampling. The first is an exact sampling algorithm based on dynamic programming with time and space exponential in $k$. The second is an approximate sampling algorithm based on Markov chains with a running time that is $\text{poly}(k)$.

### 3.1   Exact sampling with a dynamic program

First note that the sampling problem is easily solvable in time $O(n^k + k \log k)$. Indeed, we can enumerate all the top-$k$ lists (there are at most $n^k$ of them) and for each list $\tau$, in time $O(k \log k)$, compute $K^{(p)}(\tau)$. This completely describes the distribution on $T_{k,n}$ and it is easy to sample exactly from this distribution in time $O(n^k + k \log n)$.

We now present a more efficient exact sampling algorithm with running time exponential in $k$ but logarithmic in $n$. The main intuition behind the algorithm is that two top-$k$ lists that differ only by elements in $\{k+1, \ldots, n\}$ have the same probability of being sampled. In particular we can decompose the Kendall distance between a top-$k$ list $\tau$ and the identity $I_k$ by considering separately the number of inversions in the first $k$ elements, the number of inversions between the first $k$ and the last $n - k$ elements, and the number of elements in $k$ that are incomparable. All the top-$k$ lists for which these three quantities are the same are equiprobable.

We now formally present this approach. Let $\ell \in \{1, \ldots, k\}$ and let $P, Q \subseteq [k]$ such that $|P| = \ell = |Q|$. Let $m \in \{0, 1, \ldots, \binom{k}{2}\}$. Define $T_{k,n}(P, Q, m) \subseteq T_{k,n}$ to be the following set. A top-$k$ list $\tau \in T_{k,n}(P, Q, m)$ iff (i) $[k] \cap \tau = P$; (ii) the elements of $P$ occur only in positions given by $Q$; and (iii) the contribution to $K^{(p)}$ by the elements in $P$ is $m$, i.e.,

$$\sum_{i,j \in [k] \cap \tau \text{ and } i < j} \bar{K}^{(p)}_{i,j}(\tau) = m.$$

First we show that each of the partitions consists of equiprobable top-$k$ lists.

**Lemma 1.** *For any two top-$k$ lists $\tau, \tau' \in T_{k,n}(P, Q, m)$, it holds that $K^{(p)}(\tau) = K^{(p)}(\tau')$.*

Using this, we present our first sampling algorithm.

**Theorem 2.** *There is an algorithm that generates a top-$k$ list according to $\mathcal{M}^{(p)}_{\beta,k,n}$ in time $O(k^2 4^k + k^2 \log n)$.*

## 3.2 Approximate sampling with a Markov chain

In this section, we present methods based on Markov chains to approximately sample from $\mathcal{M}_{\beta,k,n}^{(p)}$. Let $\mathcal{C}$ be a Markov chain on a state space $\Omega$. We use $\mathcal{C}(i,j)$ to denote the transition probability from state $i \in \Omega$ to $j \in \Omega$. In our case, the chains will be defined on $\Omega = T_{k,n}$ and will be ergodic, guaranteeing convergence to a unique stationary distribution. Recall that there are different measures to analyze the speed of a Markov chain's convergence to stationarity. Here we analyze the *relaxation time*, which is defined as $t_{\mathrm{rel}} = 1/(1 - \lambda_2)$ where $\lambda_2$ is the second largest eigenvalue of the transition probability matrix of the chain. It is usually more convenient to analyze the relaxation time of a Markov chain when the stationary distribution is hugely biased. See [19] for formal definitions and background on Markov chains.

Our goal here is to construct Markov chains whose relaxation time is small. We construct a chain and show a relaxation time bound of $O(k^5 \log k)$ for it.

Let $\tau^* \in T_{k,n}$ be the center. We define the Markov chain $\mathcal{C}$ on $T_{k,n}$ converging to $\mathcal{M}_{\beta,\tau^*}$. We say two elements $x_i$ and $x_j$ to be $\tau^*$-*adjacent* if there is no $x_k$ such that $x_i >_{\tau^*} x_k >_{\tau^*} x_j$ or $x_j >_{\tau^*} x_k >_{\tau^*} x_i$.

**Definition 1** (Chain $\mathcal{C}$)**.** *Let $\tau \in T_{k,n}$. Choose $1 \leq i \leq k-1$ u.a.r. and equiprobably do one of:*

*(i)* Transposition step: *Equiprobably do one of:*

> $T_0$*: If $\tau(i) \in \tau^*$, then find minimum $j > i$ such that $\tau(j) \in \tau^*$ and put them in the order of $\tau^*$ w.p. $e^\beta/(1 + e^\beta)$ and the opposite order w.p. $1/(1 + e^\beta)$.*
> $T_1$*: If $\tau(i) \notin \tau^*$, then find minimum $j > i$ such that $\tau(j) \notin \tau^*$ and put them in the order of $\tau^*$ w.p. $1/2$ and the opposite order w.p. $1/2$.*
> $T_2$*: If ($\tau(i) \in \tau^*$ and $\tau(i+1) \notin \tau^*$) or ($\tau(i) \notin \tau^*$ and $\tau(i+1) \in \tau^*$), then put them in the order of $\tau^*$ w.p. $e^\beta/(1 + e^\beta)$ and the opposite order w.p. $1/(1 + e^\beta)$.*

*(ii)* Substitution step: *W.p. 1/2 stay at the current state, and w.p. 1/2 equiprobably do one of:*

> $S_0$*: A homogeneous substitution, i.e.,*
>> $S_{00}$*: If $\tau(i) \in \tau^*$, then let $x_i$ be the $\tau^*$-adjacent element such that $x_i >_{\tau^*} \tau(i)$ and if $x_i \notin \tau$, then replace $\tau(i)$ by $x_i$ w.p. $e^\beta/(1 + e^\beta)$. If $\tau(i) \in \tau^*$, then let $x_j$ be the $\tau^*$-adjacent element such that $x_j <_{\tau^*} \tau(i)$ and if $x_j \notin \tau$, then replace $\tau(i)$ by $x_j$ w.p. $1/(1 + e^\beta)$.*
>> $S_{01}$*: If $\tau(i) \notin \tau^*$, pick $c$ u.a.r. from $\bar{\tau}$ and if $c \notin \tau^*$, replace $\tau(i)$ by $c$ w.p. $1/2$.*
> $S_1$*: A non-homogeneous substitution, i.e., choose $c$ u.a.r. from $\bar{\tau}$ and compare it with $\tau(k)$ and if one of them is in $\tau^*$ and the other one is not, keep the $\tau^*$ element inside w.p. $e^{\beta(1+p\cdot i)}/(1 + e^{\beta(1+p\cdot i)})$ and the element outside $\tau^*$ w.p. $1/(1 + e^{\beta(1+p\cdot i)})$; here, $i = |\tau[1, k-1] \cap \tau^*|$.*

*If the premise is not satisfied in any of the above, then do nothing.*

We first show that the stationary distribution $\Pi$ of $\mathcal{C}$ is the desired top-$k$ Mallows distribution.

**Lemma 3.** $\Pi = \mathcal{M}_{\beta,\tau^*}$.

We next bound the relaxation time of $\mathcal{C}$. To do this, we employ a useful technique from Markov chains known as *decomposition*. In this technique, we partition the Markov chain $\mathcal{C}$ into smaller Markov chains $\mathcal{C}^{(1)}, \ldots, \mathcal{C}^{(k)}$ and connect the $\mathcal{C}^{(i)}$'s by another Markov chain $\bar{\mathcal{C}}$. It follows that $t_{\mathrm{rel}}(C) \leq t_{\mathrm{rel}}(\bar{\mathcal{C}}) \cdot \max_i t_{\mathrm{rel}}(\mathcal{C}^{(i)})$; see [22]. Here, we partition the state space into $k+1$ parts. Indeed, for $0 \leq i \leq k$, let $T_{k,n}^{(i)} = \{\tau \in T_{k,n} \mid |\tau \cap \tau^*| = i\}$. We define the restriction of $\mathcal{C}$ to each partition $T_{k,n}^{(i)}$ as follows: the chain $\mathcal{C}^{(i)}$ performs exactly like $\mathcal{C}$ in Definition 1 except it never takes the $S_1$ option. Let $\bar{\mathcal{C}}$ denote the chain that connects these partitions; we will present its transition probabilities and bound $t_{\mathrm{rel}}(\bar{\mathcal{C}})$ later.

To present the analysis, we also need two additional concepts. The first concept concerns biased and unbiased adjacent transposition chains. Given an $n \times n$ non-negative matrix $P(\cdot, \cdot)$ with entries in $[0, 1]$, an $n$-*adjacent transposition* Markov chain is defined on $S_n$ as follows: at state $\tau \in S_n$, pick $i$ u.a.r. and place $\tau(i)$ and $\tau(i+1)$ in the natural order w.p. $P(\tau(i), \tau(i+1))$ and in the opposite order w.p. $1 - P(\tau(i), \tau(i+1))$. We call the chain *unbiased* if the entries of $P$ are $1/2$ and *biased* otherwise. If there is a constant $p > 1/2$ such that $\forall j > i, P(i, j) = p$, then the relaxation time of a

biased adjacent transposition chain is $n^2$ [2]. For the unbiased chain the relaxation time is bounded by $n^3 \log n$ [25]. (Notice that the mixing time of the biased chain is independent of $p$. This is quite common when analyzing the mixing time of random walks, e.g., a biased random walk on a line.) [2]

The second concept concerns exclusion processes. Given integers $k, n$ such that $k < n$ and a $p \in [0, 1]$, an $(n, k, p)$-*exclusion process* is a Markov chain defined on the subset of $n$-bit strings of Hamming weight $k$ that makes the following transpositions: at a state $x$ choose $i \in [n]$ u.a.r.; if $x(i) = x(i + 1)$, then the chain stays at state $x$, otherwise the new state has $(x(i), x(i + 1) = (1, 0))$ w.p. $p$ and $(0, 1)$ w.p. $1 - p$. The relaxation time of this exclusion process is $n^2$ [2].

**Lemma 4.** *For each $i \in [k]$, $t_{\mathrm{rel}}(\mathcal{C}^{(i)}) = O(k^3 \log k)$.*

We now proceed to define $\bar{\mathcal{C}}$. This will be a chain defined on the partitions, i.e., its sample space will be $\{T_{k,n}^{(i)}\}$ and the transition will be defined by the $S_1$ step in Definition 1. Let $q_i = \Pr_{\tau \in T_{k,n}^{(i)}}[\tau(k) \notin \tau^*]$. Note that $q_i$ is decreasing with respect to $i$: $q_1 = 1 - 1/Z$ and $q_{k-1} = e^{(k-2)\beta}/Z$, where $Z = \frac{e^{(k-1)\beta} - 1}{e^\beta - 1} \leq 2e^{(k-2)\beta}$.

From state $T_{k,n}^{(i)}$, the chain moves to $T_{k,n}^{(i+1)}$ w.p. $r_i = q_i e^{\beta(1+p(k-i))}/(1 + e^{\beta(1+p(k-i))})$, moves to $T_{k,n}^{(i-1)}$ w.p. $\ell_i = (1 - q_i)/(1 + e^{\beta(1+p(k-i))})$, and does nothing w.p. $1 - r_i - \ell_i$. Note that since $q_i \geq 1/2$, we always have $r_i > 1/2$. Furthermore we always have $r_i/\ell_{i+1} > 1$.

**Lemma 5.** $t_{\mathrm{rel}}(\bar{\mathcal{C}}) \leq k^2/4$.

Finally, we are ready to bound the relaxation time of $\mathcal{C}$. Using Lemma 4, Lemma 5, and employing the decomposition technique, we obtain the following:

**Theorem 6.** $t_{\mathrm{rel}}(\mathcal{C}) = O(k^5 \log k)$.

We remark here that using the above result, we can also bound the total variation mixing time, which is another measure for mixing. The *total variation mixing time*, denoted by $t_{\mathrm{tv}}$, is defined as the minimum time after which the $L_1$ distance of the current state of $\mathcal{C}$ and $\Pi$ falls below a constant, say, $1/4$. From Theorem 6, and the fact that $t_{\mathrm{rel}} \leq t_{\mathrm{tv}} \leq t_{\mathrm{rel}} \log(\min_x 1/\Pi(x))$ we conclude that $t_{\mathrm{tv}} \leq \beta k^7 \log k$. In the supplementary material we show $t_{\mathrm{tv}} = \Omega(k^3)$. Our experimental results suggest that the $L_1$ distance falls bellow a constant in almost $k^3$ number of steps.

## 4 Reconstruction

In this section we focus on two basic learning questions in the spirit of [1, 5, 6, 8]. How many samples from the top-$k$ Mallows model do we need to observe to reconstruct the center top-$k$ list? And, given samples from a mixture of several top-$k$ lists, how to learn the individual components of the mixture.

### 4.1 Learning the center

In this section we give a simple algorithm for provably reconstructing the central top-$k$ list, given enough samples from $\mathcal{M}_\beta$. The main idea is to track the ordering among pairs of elements generated by $\mathcal{M}_\beta$ and use this information to reconstruct the center. For the remaining of this section, whenever $\tau \in T_{k,n}$ is a random variable, we assume it is generated according $\mathcal{M}_{\beta,\tau^*,k,n}$. For simplicity, we assume that the center $\tau^* = I_k$, the identity top-$k$ list.

First we bound the probability of an inversion a top-$k$ Mallows model.

**Lemma 7.** *For any two elements $i < j$, $i \in [k]$, it holds that $\Pr_\tau[i <_\tau j] \geq \exp(\beta) \cdot \Pr_\tau[j <_\tau i]$.*

We next bound the probability that the top-$k$ Mallows model contains a given element.[3]

**Lemma 8.** *For any $i \in [k]$, it holds that $\Pr_\tau[i \in \tau] = \Omega(\exp(\beta)/(n-k))$.*

Using these two bounds, we finally obtain the center reconstruction algorithm.

**Theorem 9.** *There exists a polynomial time algorithm that uses $\Theta\left(\frac{e^{-\beta}+1}{e^\beta-1}(n-k)\log n\right)$ samples from $\mathcal{M}_\beta$ and can identify the central top-$k$ list with probability $1 - o(n^{-3})$.*

## 4.2 Learning mixtures

In this section we consider the problem of learning a uniform mixture of several top-$k$ Mallows models. Let $\tau_1^*, \ldots, \tau_t^*$ be the centers and let $\beta$ be the common decay parameter. For simplicity of exposition we assume in this section that $\beta \le 1$, i.e., we assume to be in the case where samples can end up being far from the center. Note that this assumption is not crucial and similar bounds can be derived for $\beta > 1$. In the uniform mixture model, first an $i \in [t]$ is chosen u.a.r. and a top-$k$ list is generated according to $\mathcal{M}_{\beta,\tau_i^*}$. The goal of the reconstruction problem is, given enough samples generated according to the mixture, learn their centers.

The main idea behind the algorithm is to first cluster the given samples into $t$ clusters and then apply the center reconstruction to each cluster. To be able to do the first step without any error, we need a simplifying assumption that every pair of centers is sufficiently far apart.

First, we observe that samples from a top-$k$ Mallows model will mostly end up close to its center. For simplicity of exposition, we introduce a slight abuse of notation: if $\tau$ is a top-$k$ list, we will sometimes use $\tau$ as if it were the set of its top-$k$ elements. Thus $\tau \cap \tau'$ will denote the set of elements that are within the first $k$ of both $\tau$ and $\tau'$, and $\tau \setminus \tau'$ will denote the elements that are both within the first $k$ of $\tau$ but not within the first $k$ of $\tau'$.

**Lemma 10.** *If $\beta \le 1$, then we have $\Pr_{\tau \sim \mathcal{M}_{\beta,\tau^*}}\left[|\tau \cap \tau^*| \le k - \sqrt{\beta^{-1}3k\ln n}\right] < n^{-2k}$.*

Given this, provided that the centers in the mixture are sufficiently far from each other [4], we can cluster sampled top-$k$ lists, using single-linkage clustering in such a way that two samples are clustered together if and only they were produced by the same center.

**Lemma 11.** *Suppose that $\beta \le 1$ and, for each $\{i,j\} \in \binom{[t]}{2}$, it holds $|\tau_i^* \setminus \tau_j^*| > 4\sqrt{\beta^{-1}3k\ln n}$ and suppose each $\mathcal{M}_{\beta,\tau_i^*}$ contributes at most $o(n^{2k})$ samples. Then, we can cluster sampled top-$k$ lists in polynomial time so that, w.p. $1 - o(1)$, two samples end up in the same cluster if and only if they were generated by the same center.*

Using the clustering, the algorithm for mixtures is easy.

**Theorem 12.** *Suppose that $\beta \le 1$, that $t = o\left(\frac{\beta \cdot n^{k-1}}{\log n}\right)$ and that, for each $\{i,j\} \in \binom{[t]}{2}$, it holds $|\tau_i^* \setminus \tau_j^*| > 4\sqrt{\beta^{-1}3k\ln n}$. Then, there is an algorithm that uses*

$$O\left(t \cdot \frac{e^{-\beta}+1}{e^\beta-1}(n-k)\log(nt)\right) \le O\left(\beta^{-1} \cdot nt \cdot \ln(nt)\right),$$

*samples to identify the $t$ central top-$k$ lists with probability $1 - o(n^{-3})$.*

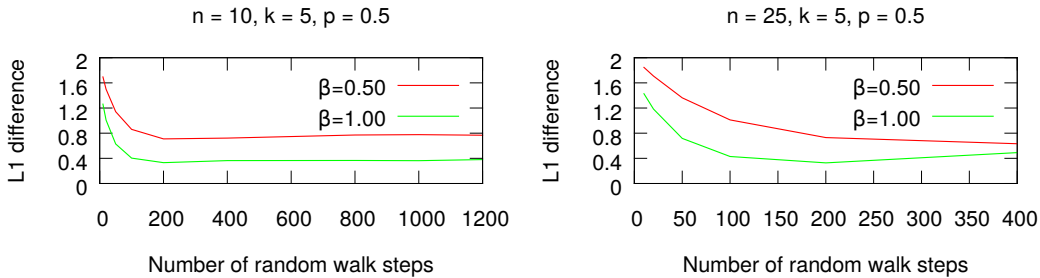

Figure 1: Empirical performance of the Markov chain based approximate sampling. The two plots correspond to $n = 10$ (samples=10K) and $n = 25$ (samples=1M), both having $p = 0.5$.

[4]An oft-made assumption in provable learning of mixtures.

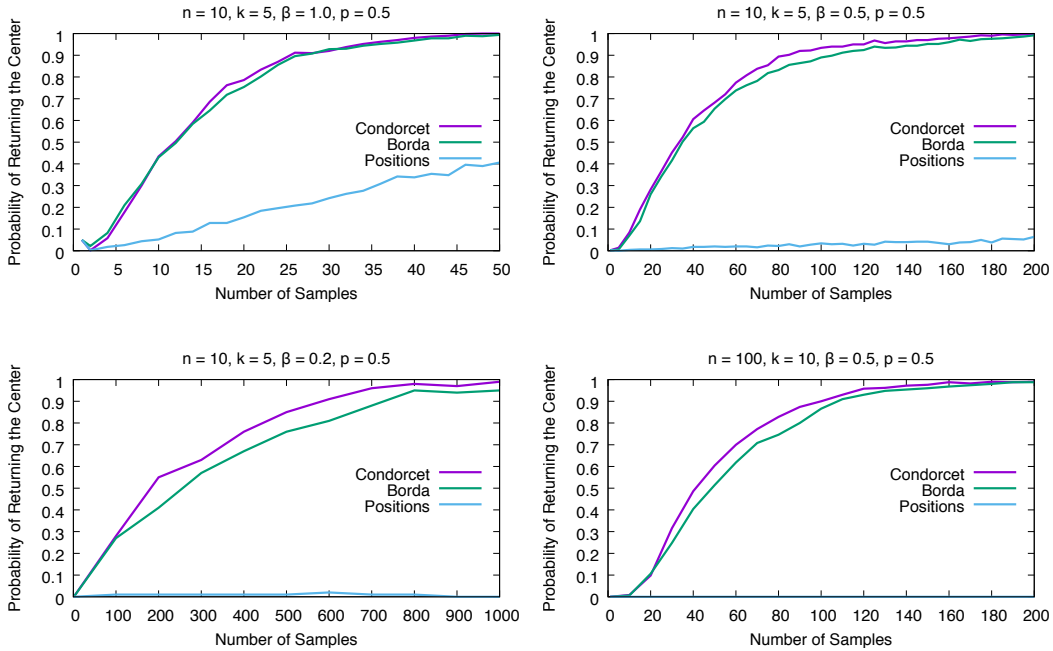

Figure 2: The first three plots detail the results for $n = 10$ and $k = 5$, $p = 0.5$, and $\beta \in \{0.2, 0.5, 1.0\}$. The last plot, instead, has $n = 100$ and $k = 10$, and $p = \beta = 0.5$. In each of the plots, the $y$-axis has range $[0, 1]$ and represents the probability that the algorithms return the correct center; the ranges of the $x$-axis, which represents the number of samples, vary across the plots.

## 5  Experiments

We describe two sets of experiments. We first show that our Markov chain based algorithm can efficiently sample top-$k$ lists. We then use this sampler to create synthetic datasets from known centers and show the applicability of our reconstruction results. While our reconstruction algorithm from Theorem 9 assumes that there are enough samples, we generalize it based on the Condorcet criterion and a tie-breaking rule. We show that our proposed algorithm outperforms reasonable baselines on synthetic datasets. We apply this proposed algorithm on a real dataset of top-$k$ movie lists to create an aggregate list and present anecdotal evidence.

**Approximate sampling.** First we test the convergence of the Markov chain sampling algorithm (Section 3.2). We start a number of walks from a given center and conduct walks for a fixed number of steps. We calculate the empirical distribution of the endpoints and then compute the $L_1$ difference of this empirical distribution from the true distribution (calculated via the dynamic program in Section 3.1). This $L_1$ difference is then plotted against the number of steps used by the random walk. In Figure 1 we plot the results for $n = 10$, $k = 5$ and $n = 25$, $k = 5$. Each plot contains two lines, one for each value of $\beta$ in $(0.5, 1.0)$, $p$ being set to $0.5$ all through. For $n = 10$, the empirical distribution is computed using $10K$ different walk samples and $1M$ for $n = 25$. Note that the state space increases as $O(n^k)$ and hence a larger value of $n$ needs substantially more samples in order to achieve similar $L_1$ distances. Also, since a smaller value of $\beta$ results in a more uniform distribution over the state space, the $L_1$ difference of the random walk sampler for $\beta = 0.5$ is higher than the one for $\beta = 1.0$. In each of the cases, we see the $L_1$ difference stabilizing after roughly 200 steps, any fluctuation after that is due to variance. In each case, it stabilizes to a non-zero value, as the number of samples in each case only a constant fraction of the support, and this does not let the $L_1$ difference between the empirical distribution and the true one to go to zero.

**Center reconstruction.** Next we study the problem of center reconstruction for top-$k$ lists.

(i) We propose to extend our algorithm from Theorem 9 to the case when are not enough samples by using the Condorcet criterion [3] as follows. Given the set of input top-$k$ lists, for the generic pair of elements $\{i, j\}$, we say that $i$ (resp., $j$) beats $j$ ($i$) in a head-to-head contest if the number of top-$k$ lists where $i > j$ ($j > i$) is larger than the number of top-$k$ lists where $j > i$ ($i > j$); if the two

numbers are same, then we say that the contest is tied. The *value* of $i$ is defined to be the number of $j \neq i$ such that $i$ beats $j$ in a head-to-head contest. Our proposed algorithm orders the elements according to their value; if the first $k + 1$ elements have no value-ties, then the algorithm returns the first $k$ elements in the ordering. It is easy to show that, if the condition of Theorem 9 is satisfied, then with high probability there will be no value-ties and the returned top-$k$ list will be center. We also implemented a tie-breaking rule[5] if the number of samples is not large enough.

(ii) The Borda count algorithm (named after the Borda method [3]), given a top-$k$ list, assigns a score of $k$ to the topmost element of the list, a score $k - 1$ to the second element of the list, down to a score of 1 to the $k$th element of the list, and 0 to every other element. The elements are sorted in a decreasing order according to the sum of its scores across the input top-$k$ lists. If there are no ties in the first $k + 1$ elements, the algorithm returns the first $k$ elements and declares failure if otherwise.

(iii) As another baseline, we consider the following positional algorithm. The algorithm assigns to each element its most frequent position in the input top-$k$ lists. The ideal position for the generic element $i$ is its most frequent position in the range $\{1, \ldots, k\}$; if there are ties, the algorithm fails. If, for each $\ell \in [k]$, there exists exactly one element whose ideal position is $\ell$, then the returned top-$k$ list will be the perfect matching between the positions and the elements.

We ran the algorithms on synthetic datasets, as well as on a real world dataset. The synthetic datasets were all generated by the top-$k$ Mallows model, having a single center.

*Synthetic datasets.* We first produced a number of synthetic top-$k$ lists by sampling from $\mathcal{M}^{(p)}_{\beta, I_k, k, n}$, for a number of choices of $n, k, \beta, p$, by running the Markov chain (Section 3.2) for 1000 steps. Figure 2 shows the probability of returning the center (averaged over various runs) of the algorithms on these inputs. The Condorcet algorithm clearly outperforms the other two.

*Criterion dataset.* This dataset contains the top-10 movies lists of a number of directors, actors and artists, as collected by the Criterion web site. We acquired 176 movie lists from `www.criterion.com/explore/top10` and truncated each list to the first 10 movies. The central top-10 list returned by the Condorcet algorithm ran on the aforementioned 176 top-10 lists is, starting from its top element: *Gilliam's* "Brazil", *Fellini's* "8 ½", *Reed's* "The Third Man", *Maysles, Maysles & Zwerin's* "Gimme Shelter", *Kurosawa's* "Shichinin no Samurai", *Laughton's* "The Night of the Hunter", *Bresson's* "Au hasard Balthazar", *Fassbinder's* "Angst essen Seele auf", *Pontecorvo's* "La battaglia di Algeri", *Carné's* "Les Enfants du Paradis". Most of the movies in this list are considered to be masterpieces of the 7th Art.

# 6    Conclusions

In this work we proposed a top-$k$ Mallows model that generalizes the classic Mallows model to top-$k$ lists. While our model is apparently challenging from an analysis point of view, we show that it is still possible to design efficient and provably good algorithms for sampling and reconstruction. Our work opens several promising research directions, including improving the running times, sample complexity bounds, and extending the model to other measures on top-$k$ lists [7].

### Acknowledgments

Flavio Chierichetti and Shahrzad Haddadan were supported by the ERC Starting Grant DMAP 680153, by the SIR Grant RBSI14Q743, by a Google Focused Award, and by the "Dipartimenti di Eccellenza 2018-2022" grant awarded to the Dipartimento di Informatica at Sapienza.

## Footnotes

[1]In fact, different $p$ values capture different scenarios. For example, if the top-$k$ list is the most livable cities, a bigger $p$ might be desirable, whereas if the top-$k$ list presents movies of a certain genre, smaller $p$ might be appropriate.

[2]In fact, ours is a random walk on a partial order: $\sigma \leq \tau$ iff when going from $\sigma$ to $\tau$ the number of transpositions from placing a bigger number ahead of a smaller number is greater than the number of transpositions from placing a smaller number ahead of a bigger number. Similar to what happens for a walk on a total order where a particle moves to left w.p. $q$ and to right w.p. $1 - q$, the mixing time is maximized when $q = 1/2$.

[3]For simplicity we provide only a bound that involves only involving only $\beta$, $n$ and $k$, although we note a tighter bound could be obtained involving by $p$ as well.

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
