[Supplementary Material]

# Mallows Models for Top-$k$ Lists
# (Supplementary material)

**Flavio Chierichetti**
Sapienza University, Rome, Italy
flavio@di.uniroma1.it

**Anirban Dasgupta**
IIT, Gandhinagar, India
anirban.dasgupta@gmail.com

**Shahrzad Haddadan**
Sapienza University, Rome, Italy
shahrzad.haddadan@uniroma1.it

**Ravi Kumar**
Google, Mountain View, CA
ravi.k53@gmail.com

**Silvio Lattanzi**
Google, Zurich, Switzerland
silviolat@gmail.com

## A   Proofs for Section 3.1

### A.1   Proof of Lemma 1

*Proof.* We partition the set $\{(i,j) \subseteq [k] \cup \tau, i > j\}$ of pairs into the following nine disjoint subsets of the form $S_{A,B} = \{(i,j) \mid i > j, i \in A, j \in B\}$, where $A, B \in \{[k] \setminus \tau, P, \tau \setminus [k]\}$. Let $\kappa(S_{A,B})$ be the contribution of $S_{A,B}$ to $K^{(p)}$.

Since $i > j$, it is easy to see that $S_{\tau \setminus [k],[k] \setminus \tau} = S_{\tau \setminus [k], P} = \emptyset$ and $\kappa(S_{P,[k] \setminus \tau}) = 0$. Furthermore, by calculations, we obtain $\kappa(S_{[k] \setminus \tau,[k] \setminus \tau}) = \kappa(S_{\tau \setminus [k], \tau \setminus [k]}) = p\binom{k-\ell}{2}$, $\kappa(S_{[k] \setminus \tau, \tau \setminus [k]}) = (k - \ell)^2$, and $\kappa(S_{P,P}) = m$, the number of inversions in $P$.

Finally, $\kappa(S_{[k] \setminus \tau, P})$ can be calculated by summing for each $j \in P$, the number of $i > j$ such that $i \notin P$. Similarly, $\kappa(S_{P,\tau \setminus [k]})$ can be calculated by summing for each position $x \in Q \subseteq \tau$, the count of non-$Q$ positions above it. Note that both these numbers can be calculated since the sets $P$ and $Q$ have been fixed. This proves the claim. $\qquad\square$

### A.2   Proof of Theorem 2

*Proof.* Note that the number of partitions, $T_{k,n}(P,Q,m)$, is bounded by
$$\sum_{\ell=0}^{k} \binom{k}{2}\binom{k}{\ell}\binom{k}{k-\ell} = \binom{k}{2}\sum_{\ell=0}^{k}\binom{k}{\ell}^2 \le O(k^2 4^k).$$
The algorithm first starts by counting the number of top-$k$ lists in each of the $T_{k,n}(P,Q,m)$ for all legitimate values of $P$, $Q$, and $m$. For a particular $T_{k,n}(P,Q,m)$, the total number of top-$k$ lists can be counted in the following manner. There are exactly $\binom{n}{k-\ell}$ ways to fill up the positions in $Q$. The total number of top-$k$ lists in this partition is a product of $\binom{n}{k-\ell}$ with the number of ways of ordering the elements of $P$ such that they have exactly $m$ inversions. Counting the number of lists of size $\ell$ that have exactly $m$ inversions can be done using the following standard dynamic program. Indeed, wlog, we can consider $P$ to be the set $\{1, \ldots, \ell\}$. Let $M$ be a table of size $\ell \times \binom{\ell}{2}$, where $M[x,y]$ indicate the number of lists of size $x$ that have $y$ inversions. Initialize $M[1,0] = 1$ and $M[1,z] = 0$ for all $z > 0$. Since the list of size $x$ can be obtained by placing the last element in $0, 1, \ldots x - 1$, define $M[x,y] = \sum_{z=\max(0,y-x+1)}^{y} M[x-1,z]$.

The sampling algorithm proceeds by first choosing one of the $T_{k,n}(P,Q,m)$ partitions proportional to its count. We then sample a top-$k$ list $\tau$ in the chosen partition uniformly at random. To do so, we first sample the $|\bar{Q}|$ elements in $\tau$ from $\{k+1, \ldots, n\}$ in time $O(\log\binom{n}{k})$. Next, we need to sample

a random ordering of the elements of $P$ with exactly $m$ inversions. To do it we use the matrix $M$ defined above. We start by placing the last element in position $z'$ with probability proportional to $M[\ell - 1, \ell - z']$, then the second last element in position $z''$ between the available positions (i.e., not considering the position already taken by element $\ell$) with probability $M[\ell - 2, \ell - 1 - z'']$, and so on. By a simple induction we can see that the permutation constructed in this way is uniform at random (u.a.r.) among the set of permutations of $P$ with $m$ inversions, as in each step we restrict to a subset of permutations with probability proportional to the size of the subset.

Thus overall we can sample a random top-$k$ list in time $O(k^2 4^k + k \log n)$. The correctness of the sampling procedure follows from Lemma 1 as well as from the above claim of sampling u.a.r. from each of the partitions. $\qquad\square$

# B  Proofs for Section 3.2

## B.1  Proof of Lemma 3

*Proof.* We show for any two $\tau_1, \tau_2 \in T_{k,n}$ the detailed balanced equation holds, i.e., $\Pi(\tau_1)\mathcal{C}(\tau_1, \tau_2) = \Pi(\tau_2)\mathcal{C}(\tau_2, \tau_1)$. We verify the condition when $\tau_1$ and $\tau_2$ are reachable from each other by $S_1$; the other cases of $T_0, T_1, T_2, S_0$ are similar (omitted). Consider $\tau_1, \tau_2$ such that $\tau_1(j) = \tau_2(j)$ for $j < k$, $\tau_1(k) = c \in \tau^*$, and $\tau_2(k) = c' \notin \tau^*$ and $c' \in \bar{\tau}_1$. By swapping $c$ and $c'$, the following inversions will be added to $\tau_1$: for any $x \notin \tau^*$ in $\tau_1[1, k-1]$, we have $x \parallel_{\tau_2} c'$ and $x >_{\tau^*} c'$; furthermore, $c <_{\tau_2} c'$ and $c >_{\tau^*} c'$. Thus, $K^{(p)}(\tau_2) - K^{(p)}(\tau_1) = i \cdot p + 1$ where $i$ is $|\tau_1[1, k-1] \cap \bar{\tau^*}|$, and $\Pi(\tau_1)/\Pi(\tau_2) = e^{\beta(i \cdot p + 1)}$. Hence we have $\Pi(\tau_1)/\Pi(\tau_2) = \mathcal{C}(\tau_2, \tau_1)/\mathcal{C}(\tau_1, \tau_2)$. $\qquad\square$

## B.2  Proof of Lemma 4

*Proof.* The key observation is that $\mathcal{C}^{(i)}$ is the product of the following Markov chains, one corresponding to each option in Definition 1 and each of whose relaxation times are known or can be easily analyzed. $\mathcal{C}_{T_0}$ is a biased $i$-adjacent transposition Markov chain on the symmetric group $S_i$; $\mathcal{C}_{T_1}$ is an unbiased $(k-i)$-adjacent transposition Markov chain on $S_{k-i}$; $\mathcal{C}_{T_2}$ is a $\left(k, k-i, 1/(1+e^\beta)\right)$-exclusion process; $\mathcal{C}_{S_{00}}$ is a $\left(k, i, 1/(1+e^{\beta p})\right)$-exclusion process; and $\mathcal{C}_{S_{01}}$ is an instance of the coupon collector problem.

Appealing to the relaxation time bounds for biased and unbiased adjacent transposition Markov chains, and exclusion processes, the proof follows from a result [1] that relates the relaxation time of several Markov chains to their product. $\qquad\square$

## B.3  Proof of Lemma 5

*Proof.* In this proof we bound the conductances of $\mathcal{C}$. Recall that for a Markov chain on $\Omega$ with transition probability $P$ and stationary distribution $\Pi$, and for a subset $S \subseteq \Omega$ the conductance of set $S$ is defined by $\Phi_S = \Pi(S)^{-1} \sum_{x \in S, y \notin S} \Pi(x)P(x, y)$. The conductance of the Markov chain $C$ is defined as $\Phi_C = \min_{S; \Phi(S) \leq 1/2} \Phi_S$, and the relaxation time is related to it by the Cheeger inequality: $t_{\text{rel}} \leq 1/\Phi^2$. We prove a more general statement (Lemma B.1 below) from which we conclude $\Phi_\mathcal{C} > 1/2k$, which completes the proof. $\qquad\square$

**Lemma B.1.** *Consider a lazy Markov chain defined on a path of length $k$ where the vertices are indexed by $V = \{0, 1, \ldots, k-1\}$, and with the following transition probabilities: for any $0 \leq i \leq k-1$, $P(i, i+1) = r_i$ and for any $1 \leq i \leq k$, $P(i, i-1) = \ell_i$. If these probabilities satisfy: for all $1 \leq i \leq k-1$, $r_i/\ell_{i+1} \geq 1$, and $r_i > p$ then the conductance of this Markov chain is at least $p/k$.*

*Proof.* Let $\Pi$ be the stationary distribution of this chain. Since $r_i/\ell_{i+1} \geq 1$ for all $i$, we have $\Pi(j) \geq \Pi(i)$ for any $j > i$. Now consider a subset $S \subseteq V$ and let $j$ be the vertex with the largest index in $S$. If $j \neq k$, then we can get out of $S$ w.p. $r_j > p$, since size of $S$ is less than $k$, and for all $i \in S, \Pi(j) \geq \Pi(i)$ and hence the conductance of $S$ is greater than $p/k$.

If the maximum index element in $S$ is $k$, then take $j$ to be the element in $S$ with maximum index such that $j - 1 \notin S$. Since $\Pi(S) \leq 1/2$, the conductance can be bounded from below: $\Phi_S \geq$

$\frac{\Pi(j)\ell_j}{\Pi(S)} \geq \frac{\Pi(j)\ell_j}{\Pi(\bar{S})} = \frac{\Pi(j)\ell_j}{\sum_{i \in \bar{S}} \Pi(i)}$, where $\bar{S}$ is the complement of $S$. Note that for any $i < j$, we have $\Pi(i)/\Pi(j) = (\Pi(i)/\Pi(j-1))(\Pi(j-1)/\Pi(j)) \leq (\Pi(j-1)/\Pi(j)) = \ell_j/r_{j-1}$ where the last inequality follows from the detailed balanced equation. Thus, $\Pi_S^{-1} \leq \frac{\sum_{i \in S^C} \Pi(i)}{\Pi(j)\ell_j} \leq kr_{j-1}^{-1} \leq kp^{-1}$. Taking the reciprocals we get the result. $\qquad\square$

## B.4  Proof of Lemma 4

*Proof.* Note that $\mathcal{C}$ can be broken into the the product of the following chains:

$\mathcal{C}_{T_0}$: A biased $i$-adjacent transposition Markov chain on the symmetric group $S_i$, denoting the relative positions of the elements in $\pi$. Hence, $t_{\mathrm{rel}}(\mathcal{C}_{T_0}) = i^2$.

$\mathcal{C}_{T_1}$: An unbiased $(k-i)$-adjacent transposition Markov chain on $S_{k-i}$, denoting the relative positions of elements in $[n] \setminus \pi$. Hence, $t_{\mathrm{rel}}(\mathcal{C}_{T_1}) = (k-i)^3 \log(k-i)$.

$\mathcal{C}_{T_2}$: A $\left(k, k-i, 1/(1+e^\beta)\right)$-exclusion process, where the $i$ zeros correspond to the positions of elements in $\tau \cap \pi$ and the $k - i$ ones correspond to the positions of elements in $\tau \cap ([n] \setminus [k])$. Hence, $t_{\mathrm{rel}}(\mathcal{C}_{T_2}) = k^2$.

$\mathcal{C}_{S_{00}}$: A $\left(k, i, 1/(1+e^{\beta p})\right)$-exclusion process where the $i$ ones correspond those elements in $\pi$ that are present in $\tau$. Hence, $t_{\mathrm{rel}}(\mathcal{C}_{S_{00}}) = k^2$.

$\mathcal{C}_{S_{01}}$: An instance of the coupon collector problem. Hence, when all the $i$ elements in $\tau \cap \pi$ are switched, this will yield a random subset and therefore $t_{\mathrm{rel}}(\mathcal{C}_{S_{01}}) = n/(n-k) \cdot k \log k$.

At this point, we appeal to a result of Diaconis and Saloff-Coste [1] that relates the relaxation time of several Markov chains to their product. Let $k \ll n$ and $p_R$ be the probability of selecting the Markov chain $R$. We have

$$
\begin{aligned}
t_{\mathrm{rel}}(\mathcal{C}^{(i)}) &= \max_{R \in \{T_0, T_1, T_2, S_{00}, S_{01}\}} \{(2/p_R) \cdot t_{\mathrm{rel}}(\mathcal{C}_R)\} \\
&= O\left(\left(\frac{n}{n-k}\right) k \log k + k^3 \log k\right) \\
&= O(k^3 \log k). \quad \square
\end{aligned}
$$

## B.5  A lower bound for $\mathcal{C}$

Let $\tau^*$ be the center of the distribution. We introduce a set $S \subset T_{k,n}$; with $\mathcal{M}_{\beta,k,n}^{(p)}(S) \geq 1/2$ such that the maximum expected time required for $\mathcal{C}$ to reach this set from an arbitrary point in $T_{k,n}$ is at least $k^3/16$.

Define $S \subset T_{k,n}$ as follows: $S = \{x \in T_{k,n}; |x \cap \tau^*| \geq k/2\}$. Clearly $\mathcal{M}_{\beta,k,n}^{(p)}(S) \geq 1/2$.

**Lemma B.2.** *Let $S = \{\tau \in T_{k,n}; |\tau \cap \tau^*| \geq k/2\}$, the expected time required for $\mathcal{C}$ to reach $S$ from $\tau^*$ is at least $k^3/16$.*

*Proof.* To reach any element in $S$ from $\tau^*$ we need to replace at least $k/2$ elements of $\tau^*$ with elements of $\bar{\tau}^*$. Let $\tau \in S$ be the first element we reach from $\tau^*$, and $\tau \cap \tau^* = \{x_1, x_2, \ldots, x_{k/2}\}$. Assume without loss of generality that the indexing is such that $x_i >_\tau x_j$ iff $i > j$. We define $X_i$ be the random variable that indicates the number of steps $\mathcal{C}$ requires until $x_i$ reaches its place in $\tau$. By linearity of expectation, the expected time to reach $\tau$ would be at least $\sum_{i=1}^{k/2} \mathbf{E}(X_i)$. Note that for $i < k/4$, $x_i$ has to pass $x_{k/4+1}, x_{k/4+2}, \ldots, x_{k/2}$, and each transposition moving $x_i$ takes place with probability $1/k$. Thus, $\mathbf{E}(X_i) \geq k^2/4$. Taking the sum over all $k/4$ elements, we get the result. $\quad\square$

## C  Proofs for Section 4.1

### C.1  Proof of Lemma 7

*Proof.* Let $S_< = \{\tau \in T_{k,n} \mid i <_\tau j\}$, $S_> = \{\tau \in T_{k,n} \mid j <_\tau i\}$. Notice that $S_< \cup S_>$ is the set of top-$k$ lists such that $\{i, j\} \cap \tau \neq \emptyset$. Define a bijection $h : S_< \to S_>$ that swaps the positions of

$i$ and $j$ in $\tau \in S_<$ to obtain $h(\tau) \in S_>$. Clearly for $\tau \in S_<$, we have $K^{(p)}(h(\tau)) \le K^{(p)}(\tau) - 1$ and hence $\Pr[\tau] \ge \exp(\beta) \cdot \Pr[h(\tau)]$. Since $h$ is a bijection, it then follows that $\Pr[\tau \in S_<] \ge \exp(\beta) \cdot \Pr[\tau \in S_>]$. $\qquad\square$

## C.2   Proof of Lemma 8

*Proof.* Fix $i \in [k]$, we partition $T_{k,n}$ based on the presence of $i$: $T_{k,n,i} = \{\tau | i \in \tau\}$. Consider the following mapping $h_i : T_{k,n} \setminus T_{k,n,i} \to T_{k,n,i}$, let $j$ be the last element in $\tau$ such that $j \notin [k]$; now, define $h(\tau)$ to be the top-$k$ list that is the same as $\tau$ but in which $i$ is replaced with $j$. For example for $n = 7$, $k = 3$, and $i = 2$, $h_2(145) = 142$. Hence $\Pr[h(\tau)] \ge \exp(\beta) \cdot \Pr[\tau]$. Furthermore, by construction, for each $\tau \in T_{k,n,i}$, $|h_i^{-1}(\tau)| \le n - k$. Thus, for $\tau' \in T_{k,n,i}$,

$$\sum_{\tau \in h_i^{-1}(\tau')} \Pr[\tau] \;\le\; e^{-\beta} \sum_{\tau \in h_i^{-1}(\tau')} \Pr[h(\tau)] \;=\; e^{-\beta} \sum_{\tau \in h_i^{-1}(\tau')} \Pr[\tau'] \;\le\; e^{-\beta}(n-k) \cdot \Pr[\tau'].$$

Applying this, we complete the proof as

$$1 \;=\; \sum_{\tau} \Pr[\tau] = \sum_{\tau' \in T_{k,n,i}} \left( Pr[\tau'] + \sum_{\tau \in h^{-1}[\tau']} \Pr[\tau] \right) \;\le\; (1 + e^{-\beta}(n-k)) \sum_{\tau' \in T_{k,n,i}} \Pr[\tau'].$$

Since $Pr_\tau[i \in \tau] = \sum_{\tau' \in T_{k,n,i}} \Pr[\tau']$, the proof is complete. $\qquad\square$

## C.3   Proof of Theorem 9

*Proof.* Suppose the algorithm samples $m$ top-$k$ lists. For every pair of elements $i$ and $j$, the algorithm decides on one of the following cases: $\{i < j, j < i, i \parallel j\}$. In order to do so, we create the count $X_{ij} = \sum_{\ell=1}^{m} X_{ij}^{\tau_\ell}$, where for each sample $\tau$, we define $X_{ij}^\tau$ to be 1 if $i <_\tau j$, $-1$ if $j <_\tau i$, and 0 otherwise. Then, for some $K > 0$, if $X_{ij} > K$, we say that $i < j$; if $X_{ij} < -K$, we say that $j < i$; and if none of the cases holds, we claim $i \parallel j$.

We show that there exists some $K > 0$, such that for each pair $i, j$, such that $i < j$, the correct decision is output. It is clear to see that this is enough to identify the original $[k]$ items as well as their correct ordering. In order to analyze the probability of correctness we define the following biased coin. Let $p = \Omega(\exp(\beta)/(n-k))$. Define $Y_1, \ldots, Y_m$ to be i.i.d random variables such that

$$Y_\ell = +1 \text{ w.p. } \frac{e^\beta p}{1 + e^\beta}; \quad -1 \text{ w.p. } \frac{p}{1 + e^\beta}; \quad 0 \text{ w.p. } 1 - p.$$

Using Lemma 7 and Lemma 8, it is clear that if $i < j$, $\Pr[X_{ij}^\tau < 1] \le \Pr[Y_\ell < 1]$. Hence, applying Bernstein's inequality [2], for $K = (1 - \epsilon)mE[Y]$ and for $0 < \epsilon < 1$, we can obtain

$$\Pr[X_{ij} < K] \le e^{-\epsilon^2 mE[Y]/2}.$$

Since $E[Y] = \frac{e^\beta - 1}{e^\beta + 1} p = \Omega\left(\frac{e^\beta - 1}{e^\beta + 1}\left(\frac{e^\beta}{n-k}\right)\right)$, by choosing $\epsilon = 0.5$, and using $m = \Theta\left(\frac{e^\beta + 1}{e^\beta - 1}\left(\frac{n-k}{e^\beta}\right)\log n\right)$ samples, the probability that the correct decision is output is at least $1 - o(n^{-3})$. By taking a union bound over all pairs, we get the stated claim. $\qquad\square$

# D   Proofs for Section 4.2

## D.1   Proof of Lemma 10

*Proof.* If $\tau$ is such that $|\tau \cap \tau^*| \le k - \sqrt{\beta^{-1} 3k \ln n}$, then $K^{(p)}(\tau, \tau^*) > 3\beta^{-1} k \ln n$, since, for each $i \in \tau^* \setminus \tau$ and for each $j \in \tau \setminus \tau^*$, we will have that $i >_{\tau^*} j$ and $i <_\tau j$, and $|\tau^* \setminus \tau| = |\tau \setminus \tau^*| \ge \sqrt{\beta^{-1} 3k \ln n}$. For each such $\tau$, we will have $\Pr[\mathcal{M}_{\beta, \tau^*} = \tau] \le e^{-3\beta^{-1} k \ln n} = n^{-3k}$. Since the number of $\tau$'s such that $|\tau \cap \tau^*| \le k - \sqrt{\beta^{-1} 3k \ln n}$ is upper bounded by the number of top-$k$ lists, i.e., by $n^k$, the proof follows from a union bound. $\qquad\square$

## D.2 Proof of Lemma 11

*Proof.* We run the following version of single-linkage clustering: start from a singleton cluster for each sample and greedily merge a pair of clusters if we find a sample $\tau'$ in one, and a sample $\tau'$ in another such that $|\tau \setminus \tau'| \le 2\sqrt{\beta^{-1}3k\ln n}$. This algorithm runs in polynomial time.

By Lemma 10, and by a union bound, the probability that at least one of the $o(n^k)$ samples $\tau$ contains in its first $k$ positions more than $\sqrt{\beta^{-1}3k\ln n}$ elements that are not in its center's first $k$ positions, is at most $o(1)$. Then, any two samples $\tau, \tau'$ generated from the same center have to satisfy $|\tau \setminus \tau'| = |\tau' \setminus \tau| \le 2\sqrt{\beta^{-1}3k\ln n}$. Therefore, w.p. $1 - o(1)$, for each center $\tau_i^*$, the samples produced by $\mathcal{M}_{\beta,\tau_i^*}$ will end up in the same cluster.

By a similar argument, w.p. $1 - o(1)$, any two samples $\tau, \tau'$ generated, respectively, by $\mathcal{M}_{\beta,\tau_i^*}$ and $\mathcal{M}_{\beta,\tau_j^*}$ for $i \ne j$ will guarantee that $|\tau \setminus \tau'| = |\tau' \setminus \tau| > 2\sqrt{\beta^{-1}3k\ln n}$. Therefore, their clusters will never be merged.

It follows that, for each center $\tau_i^*$, the final clustering will contain one cluster containing all and only the samples generated by $\mathcal{M}_{\beta,\tau_i^*}$. $\square$

## D.3 Proof of Theorem 12

*Proof.* After clustering the samples according to the Algorithm of Lemma 11, we can apply the Algorithm of Theorem 9 to compute the center of each of the $t$ clusters. $\square$