[Reviews · NeurIPS 2018]

Reviewer 1



Summary: In this paper, the authors propose an extension of the Mallows model, originally defined on the space of full rankings/permutations, to the case of top-k lists, a special kind of partial rankings. The size k of the top-k lists, additionally to the central ranking and the spread parameter, can be seen as parameter (fixed initially). They consider two problems. The first problem concerns the generation of a sample from this model. For this, they propose two algorithms (with different complexities/dependance over the whole set of items n and k). Then, they consider the problem of learning the center of such a distribution and learning a mixture of several of these distributions. To learn the center of the distributions, the proposed algorithm compares pairs of elements and decides over their order based on some threshold for the empirical pairwise probabilities. The learning mixture algorithm relies on this previous algorithms to learn the centers of the mixtures, with applying previously a single-linkage clustering step. Qualitative assessment: Clarity: The paper is well written, I understood it all reasonably well. Significance: This paper is considering an important problem and the results certainly make some contribution to the literature. Indeed, tt is true that their models builds upon/corrects some defaults of previous extensions of Mallows model to partial rankings (see l64 to l 77), since they minimize the importance of the tail elements (the one not appearing in the central top-k ranking). Their model can be also generalized to other interesting settings (see l78-91) and thus covers a diversity of rankings/applications. Their work differ from previous extensions of Mallows model on the assumptions and the distances chosen. The authors also propose several algorithms to tackle sampling/reconstruction problems. I have thus several critics. I think the bound on the relaxation time for the markov chain algorithm is still big (k^5 log(k)) even for small values of k, but the total variation mixing time is reasonable (k^3). Then, concerning the center reconstruction problem: in the proof of theorem 9, the authors propose an algorithm which order pairs of items, given that the probability of its order is below some threshold K. But in the end, in the experiments, they apply Copeland method to find the center, which is, that’s the least you can say, not a new algorithm. Then, I would have preferred some algorithms to be explained/at least sketched/have their complexities explained in the main text: the one from Theorem 2 and the one from Theorem 9. Minor comments: title: In my opinion, the title is too vague and should include "Mallow's model" l34: "most of the work has been less formal in nature » -> this formulation is a bit vague. l47: most of the pairwise relations are potentially statistically spurious: i guess that you underline the fact that some pairs of items, compared to some others, are very less observed. this may not be evident for someone non familiar with ranking. l69 to 70: the example is not very clear/ illustrative at this point l 78: typo, misses a « be » l82: why klog(n) and nlog(n)? l89: a number of our results can be generalized…-> are there limitations for the generalization? l111-113: would be more appropriate in the introduction. l161: I don’t agree on n^k. I would say n!/k!(n-k)! Proof of Lemma 8 is not very clear l271: replace o(1) with o(n^-3) experiments section: the plots are way too small Figure 1: the L1 distance is increasing for beta=1 after 200 steps, why? does the number of samples chosen in the experiments correspond to Theorem 9? l329-336: this is called Copeland method. typo in ref 19.

Reviewer 2



Summary: The authors propose a novel distance based ranking model (Mallows model) for partial ranking. More concretely, top-k rankings can be observed and the rest of the items is considered as being in the same bucket at the end of the ranked list. The authors introduce a novel distance metric which is based on Kendall distance restricted to top-k element, i.e. two items add 1 to the number of discordant pairs, if they are discordant and are in the top-k lists, they add p to the distance, if the pair of items are in the top-k only in case of one of the top-k rankings of interest, otherwise they add zero. Parameter p penalizes the event when one top-k ranking includes the pair as top ranking, whereas the other top-k ranking doesn't. The authors propose two sampling techniques, an exact one based on dynamic programing, and a second one based on MCMC. The exact seems not so practical, since large table needs to be computed and kept in the memory (as big as O(4^k)) . The MCMC based sampling seems more practical. A transition operator is designed so as the underlying chain is ergodic and the first eigengap can be computed, which, in addition, allows the authors to come up with a bound for mixing time. Reconstruction of the central top-k list is also considered for mixture model as well. The reconstruction algorithm is based on simply counting the number of wins, and if this number > K (>0) or < K or between -K and K for a pair i and j, i is considered winner, j is considered winner, or tie, respectively. Experimental study is presented to show that the proposed methods are indeed versatile. Summary of the review: I have found the topic of the paper very interesting. However, I have several concerns regarding the technical content of the manuscript, so I cannot make my decision before the author's rebuttal. Thank you for the rebuttal! I believe that this paper is ready for publication. I recommend the paper be accepted! Comments: Can the normalization factor be computed in a closed form? Regarding the MCMC part: - why the dependency on p and \beta remains hidden in the bounds? What is the reason of this? I thought that the reason of this complex transition operator is that the largest eigengap can be computed analytically, and then mixing time again can be quantified in terms of the parameters of top-k Mallows. Regarding the reconstruction part: - Braverman & Mossel 2008 and 2009 papers are based on the observation that the rank of an item coming from the model cannot differ too much'' from its ranking in the ground truth or central ranking. This property is called location stability, as far as I remember. The reconstruction technique proposed in the paper is also based on this observation, in addition to that the probability of items from the top-k list are not comparable, is also small'' which is basically the new part of the reconstruction technique. But the proposed reconstruction algorithm does not use the location stability the most efficient way, because this observation is used only to find a feasible threshold (K) for pairs of items. Whereas Braverman and Mossel came up with a bottom-up dynamic programing approach to find the center ranking which is based on the location stability too, but it makes use of this property in a more principled way than that of taking into account only pairs. It would be great to elaborate more on this issue. I mean what is similar to Braverman & Mossel's technique and what is novel. And why their DP technique cannot be applied for partial ranking. - it would be great to report the algorithms mentioned in the Theorems - in case of mixture model, assuming \beta<1 means that the random rankings (top lists) concentrates around the central top-k ranking. This is the easy case basically. Why this assumption is needed here? - the bound given in Lemma 8 should depend on p. As far as I understood, if p is small, then the model is reconstruction problem becomes very hard, because we observe items from the ground truth top-k very rarely. Regarding the experiments: - when the MCMC based sampling is tested, what is not clear to me, why the red line (\beta=0.5) converges to the green one in case of n=25 , and it doesn't in case of n=10 and k=5. \beta=0.5 is a more uniform distribution than the one with \beta=1.0, thus the L1 difference should be bigger in case if n=25 too. - when the center ranking is found, can \beta be estimated by using maximum likelihood, like in case of original Mallows? line 23: Mallows mode -> Mallows model references are not read from bib file in appendix

Reviewer 3



The authors consider the sample generation problem, center (location permutation) construction, and learning mixture model parameters for Mallows models with a top-k distance function. While most proposals for top-k lists are not invariant to tail (i.e., n-k) ordering, require generating the entire permutation, and/or analyze this from the repeated insertion model (RIM) perspective, this work defines a top-k distance function (ostensibly the distance function presented in [Klementiev, Roth & Small; ICML08] from [Fagin, et al].) and analyze the resulting top-k Mallows model in depth. For generation, exact sampling is O(k^2 4^k _ k^2 log n) and an approximate Markov chain sampler is O(k^5 log k) — which is important as even though computationally intensive, does not include n as a term, which is crucial when k is generally in the 10s and n can be in the millions. Center permutation construction and learning mixtures are also computationally efficient with provable guarantees. The Markov chain based approach for generation is used to generate synthetic data, which is then used for center permutation (re)construction, which is contrasted with Borda count and a second simple baseline — showing greater likelihood of returning the center. Finally, they consider a dataset of top-10 movie lists from the Criterion collection and return the center permutation. In summary, the problem studied in this paper is important and several theoretical findings are explained — thus, I believe it is likely that it will have some impact in the ‘rank aggregation’ community. However, the presentation and general writing is lacking, even if I think this is partially an effect of it really being a longer theoretical contribution that is compressed into conference format. Secondly, the experiments are underwhelming — the synthetic experiments perform exactly as expected and the Criterion dataset experiment doesn’t provide any meaningful evidence that this is useful. Given that there are multiple papers in this vein that have compelling experiments, a stronger paper would have empirical comparison (even possibly in terms of runtime in addition to performance) that contrasts this with the proposed method. Below is the paper evaluated with respect to the requested dimensions: [Quality]: The theoretical contribution is through and relatively compelling. By defining a different distance function, the authors provide a clever method for analysis that emits useful results and may be utilized by others for other potential structure aggregations. For example, with a suitable distance function for sequences, trees, and other structures, this is a more useful analytical framework than a RIM-style analysis. Conversely, the empirical evaluation is relatively weak. While this isn’t really the point of the paper, a stronger empirical analysis is possible and would make for a much more compelling paper (i.e., some of the examples given in the introduction). As the Criterion collection is mostly masterpieces, the resulting center permutation says almost nothing. [Clarity]: The paper is relatively clear for those who have worked in this area, but is very dense for more casual readers. For example, the ‘p-parameterized’ discussion in lines 66-70 doesn’t make intuitive sense despite them providing an example. It isn’t until one draws a picture based on line 128-134 that the example even makes sense. Secondly, while space is at a premium, the paper is largely unreadable past certain points without referring to the appendices. It would be nice if some sketches or intuitive explanation for the general analysis method could be presented. Finally, there is a general sloppiness to the paper (that could be fixed for a camera-ready). Line 4: work -> works Line 23: mode -> model Line 32: top -> top-k (twice) Line 51: [same] Line 182: section -> section, (comma) Line 284: top—k -> top-k (There are more, but this is just what I circled in the first reading) The paper can be tightened in some sections (i.e., motivation) to given intuition and improve experiments and experiment discussion. It also doesn’t help that the experiment figures are essentially unreadable without zooming in on the PDF file. [Originality]: People have clearly been thinking about top-k lists and (unless I am misunderstanding), this is the distance function from [KRS08], even if they do not analyze in nearly the same depth — as this wasn’t the point of that paper. Thus, the strength of this paper is in the execution of solving a ‘sort of known’ difficult open problem as opposed to being particularly original. [Significance]: As stated above, the theoretical results are reasonably compelling and the proof methods have potential to inspire other work. Thus, while potentially a bit esoteric, it has the potential to be an influential work. I enjoyed reading the paper and it gave me some ideas on how to analyze some of the problems I have been thinking about related to these problems. Conversely, without stronger empirical evidence, they are likely to reach a smaller audience. Overall, nice results -- but the paper needs work in general. Thus, I am inclined to 'weak reject' in its current form. Comments after author feedback: After reading the other reviews and particularly the rebuttal (and paper again), I am much more inclined to accept. This is an important problem with interesting results. I would still encourage the authors to tighten up the writing to make it more approachable (even if it is essentially a math paper).